



# Technical note: Does Multiple Basin Training Strategy Guarantee Superior Machine Learning Performance for Streamflow Predictions in Gaged Basins?

Vinh Ngoc Tran[1], Tam Van Nguyen[2], Jongho Kim[3], Valeriy Y. Ivanov[1]

[1]Department of Civil and Environmental Engineering, University of Michigan, Ann Arbor, MI 48109
[2]Department of Hydrogeology, Helmholtz Centre for Environmental Research - UFZ, 04318, Leipzig, Germany
[3]School of Civil and Environmental Engineering, University of Ulsan, South Korea

*Correspondence to*: Vinh Ngoc Tran (vinhtn@umich.edu), Valeriy Y. Ivanov (ivanov@umich.edu)

**Abstract.** In recent years, machine learning (ML) has witnessed growing prominence and popularity in hydrological science, offering convenience and ease of use without requiring extensive hydrological expertise or the complexity associated with process-based models. There exists debate regarding optimal training approaches, with some researchers advocating for multi-basin training while questioning the validity of single-basin approaches. This study examines the relationship between training dataset size (number of basins) and model performance. Through comparative analysis, we found that increasing the number of basins for ML training does not necessarily guarantee improved performance of the trained ML model. Specifically, the state-of-the-art global ML (G model) trained by Google with nearly 6,000 global basins underperforms compared to regional ML models trained with hundreds of basins in contiguous US and Great Britain regions for predicting streamflow in both gauged and ungauged basins. Furthermore, we compared the G model with our single-basin (S) ML models, trained for 609 global locations individually, and found that the G model does not consistently outperform S models, as results show S models outperforming the G model in 46% of case studies. Therefore, the training approach should not be a criterion for judging model validity; instead, the focus should be on the trained model's performance.

## 1 Introduction

With the rapid advancement of science and technology, applications of Machine Learning (ML) and Artificial Intelligence (AI) in hydrology have become increasingly popular (Ng et al., 2023; Tripathy and Mishra, 2023). Since 2018, there has been a significant surge in research publications related to ML-based streamflow forecasting, especially using Long Short-Term Memory (LSTM) networks, with multiple studies published annually demonstrating the substantial potential of novel ML architectures (Shen, 2018). Numerous studies have shown that ML can often provide higher accuracy of streamflow simulations than traditional hydrological models (Kratzert et al., 2018; Lees et al., 2021; Nearing et al., 2024; Sabzipour et al.,





2023). Machine learning models have also been applied at the global scale (Nearing et al., 2024), challenging one of hydrology's long-standing challenges: prediction in ungauged basins (PUB).

ML applications in hydrology differ fundamentally from the traditional process-based modeling approaches whose principles are based on explicit modeling of physical processes. Instead, ML focuses on aspects related to data input-output such as identifying significant input variables, optimizing data processing methodologies, fine-tuning ML architectures and hyperparameters, developing hybrid model combinations, and exploring alternative training approaches (Ng et al., 2023). The effectiveness of these ML developments is determined through simulation performance metrics computed with respect to observations, using traditional evaluation criteria such as Nash-Sutcliffe Efficiency (NSE), R-squared ($R^2$), Kling-Gupta Efficiency (KGE) coefficients, and peak error (PE) analysis (Mosavi et al., 2018; Ng et al., 2023). A common standard for evaluating quality of an ML model is based on its capability to provide reliable predictions (as evaluated with evaluation criteria) and a notable feature observed across numerous ML studies in streamflow prediction is that different case studies require different data types and input dimensions to achieve optimal model performance. However, recent discussions have emphasized the need for standardized training protocols, particularly concerning the use of data from either single (individually targeted) or multiple basins for model training. For instance, a recent study put forward two significant assertions stating (Kratzert et al., 2024):

- "*A large majority of studies that use this type of model do not follow best practices, and there is one mistake in particular that is common: training deep learning models on small, homogeneous data sets, typically data from only a single hydrological basin*"

- and "*LSTM rainfall–runoff models are best when trained with data from a large number of basins*".

where "this type of model" in the above quote refers to LSTM models trained using data from a single basin.

However, we contend that both statements are only relevant for specific applications and require further scientific evidence from other studies. Specifically, the first statement appears to be subjective as the authors do not substantiate why such a strategy should be considered flawed in a broader context: there is no comparison of their model trained using data from multiple basins versus individually trained, basic-specific models. We conjecture that the first assertion was derived from the second. While it is reasonable to assume that multi-basin training could enhance simulation performance through increased training data volume (particularly for extreme events), their conclusions appear to be valid only within a specific research context, specifically, LSTM application to the CAMELS dataset containing 531 nearly natural basins of the contiguous US (CONUS) region (Addor et al., 2017). We carry out a comparative analysis of research of single- versus multiple-basin strategy in the next section. Furthermore, it is noteworthy that approximately two-thirds of global rivers are anthropogenically influenced (cases that were not considered in their study) (Grill et al., 2019). Research conducted by Ouyang et al. (2021) demonstrated that models trained on the CAMELS dataset can perform effectively, when predicting for basins within this same dataset, but exhibit poor performance when applied to watersheds influenced by reservoirs or dams.



We contend that in fact there is no "best practice". The success of training varies on a case-to-case basis, given the availability
and quality of the data related to the governing streamflow generation mechanisms. In streamflow prediction, there are always
basins for which a trained ML model (using single or multi-basin approaches) performs poorly, with, say, NSE < 0.5 (Kratzert
et al., 2018; Ouyang et al., 2021; Nearing et al., 2024; Klotz et al., 2025). In these basins, rainfall-runoff relationships may be
influenced by other variables, not contained within a set of standard ML model inputs. If a global model combining data sets
from all commonly available records is trained, there is no guarantee that these inputs can "compensate" for the missing
variable of importance.
The field of hydrologic prediction has always been suffering from limited data and variations in their quality across regions
and time periods (Stagge et al., 2019; Mcmillan et al., 2012; Liu et al., 2017; Addor et al., 2020). In the period of active
evolution of ML applications in this field, the rigidity of "best practices" for ML training approaches should be evaluated
carefully as they inadvertently can have detrimental effects on both the continued research development (e.g., cultural
"homogenization" stifling innovation) and peer reviews of studies using alternative training approaches (e.g., reinforcement
of what constitutes a bias). While we acknowledge that models using multiple-basin ML training strategy in hydrologic
predictions can serve as a useful benchmark for comparison with a proven success in certain contexts, we contend that there
are other equally viable strategies for ML model training, questioning appropriateness of "standard" or "best practice" rules.
To substantiate the above assertion, our research presents analyses addressing two questions. Specifically, Section 2 reviews
the studies characterized as "mistakes" in Kratzert et al. (2024) and statistically evaluates whether these single-basin modeling
approaches are indeed methodologically flawed, leading to larger performance errors. Section 3 compares regional models
with global models trained by other research groups (to ensure transparency) to determine whether training with an increasing
number of basin datasets improves model performance. Finally, using the same objective we conduct a comparative analysis
between single-basin models (trained by our group) and global models trained by Google (Nearing et al., 2024).
In this study, we evaluate model performance skill using the most common metric in hydrologic predictions - the Nash-Sutcliffe
Efficiency (NSE). NSE value of 0.5 is considered as the threshold for determining whether model performance is acceptable.
This threshold magnitude was suggested by Moriasi et al. (2007) and has been widely used in numerous hydrological model
evaluation studies over the past decades. We note that this evaluation criterion has been successfully used over several decades
to assess hydrological model quality, regardless of the model's simplicity or complexity (Dawson et al., 2007; Knoben et al.,
2019; Krause et al., 2005; Smith et al., 2004; Smith et al., 2012).

**2 Previous Research Using Single-Basin Trained ML: Was It a Mistake?**

In ML applications, "mistakes" in training a ML model for streamflow prediction include: 1) model performance issues when
results are poor or below expectations; 2) input selection problems from using poor quality data instead of high-accuracy data
relevant to the study area; 3) data processing techniques that lead to issues like data leakage; and 4) Suboptimal training and



tuning approaches that prevent models from reaching optimal or near-optimal performance. These are common issues that
should be examined in most research studies.
In the first above statement, Kratzert et al. (2024) asserts that studies training a ML model (i.e., LSTM) using a single basin
are methodologically flawed, particularly highlighting 123 out of 150 papers (can be found at
https://zenodo.org/records/11247607) that used a single basin training approach across various locations across several
continents. Our review of these studies reveals that 109 of them reported NSE values (while others used KGE, R², or Root
Mean Square Error (RMSE) metrics). Focusing solely on the studies reporting NSE, we learned that 108 of them demonstrated
acceptable model performance, with NSE exceeding 0.5, 92 studies showed excellent model performance with NSE $\geq$ 0.75
(these NSE thresholds are adopted from Moriasi et al. (2007)). Only a single study reported an unsatisfactory performance
with NSE of 0.39 (Parisouj et al., 2023).
Additionally, our examination of data usage across these studies reveals that many studies used measured forcing data at local
areas instead of reanalysis/interpolated data at continental/global scales (Kim and Kim, 2021; Bai et al., 2021; Xiang and
Demir, 2021; Ouma et al., 2021; Rahimzad et al., 2021; Mao et al., 2021; Ishida et al., 2021a; Mirzaei et al., 2021; Ishida et
al., 2021b; Le et al., 2021; Li et al., 2021, just mention a few). The data processing techniques and model training approaches
show no evident flaws, particularly considering these studies have successfully passed through peer review processes.
These statistical findings demonstrate that the vast majority of studies in which a ML model was trained for individual basins
achieved their objectives with good performance, thus contradicting the claim that such a single-basin training strategy was a
"mistake".

## 3 Does Training with More Basins Improve Streamflow Predictions? Evidence from Past Research

We collected LSTM model simulation results from previously published studies conducted at both global and regional scales.
Specifically, global-scale simulation results for both gauged and ungauged basin applications were obtained from Google
(Nearing et al., 2024), where a global model was trained and tested using data from 5,680 basins. For the CONUS region,
simulation results for both gauged and ungauged basins were collected from Kratzert et al. (2024) and Kratzert et al. (2019),
wherein the gauged dataset provided results from LSTM models trained using both single-basin and multiple-basin approaches.
Simulation data for ungauged basins in the Great Britain (GB) region were obtained from Lees et al. (2021), where their model
was trained using 669 basins. The models trained with global, regional, and single basins are referred to as G, R, and S models,
respectively. To identify overlapping stream gauges across datasets, we calculated the distances between stream gauges from
the two datasets. Two stations were classified as overlapping if their spatial distance was less than one kilometer. The analysis
revealed 32 and 139 overlapping forecast locations between the global model and CONUS, and the global model and GB
datasets, respectively.



Based on the NSE (1:1) comparison shown in Fig. 1 (also Table B.1), we can conclusively address this section's question: the
LSTM model trained with data from a larger number of basins (G model) did not demonstrate enhanced performance as
compared to the R and S models. This conclusion carries greater significance as it is derived based on a comprehensive global-
scale analysis encompassing diverse case studies, a larger number of basins, and importantly, a more varied selection of
watersheds in terms of human impact on streamflow regime. It is noted that the simulation results applied for the CONUS
were collected from the same group of authors who published in previous studies, therefore ensuring fairness of comparison.
Specifically, for gauged basins, the results demonstrate that both S and R models achieved higher NSE values, as compared to
the G model at 26 and 29 locations, respectively, out of 32 total locations. Notably, these S and R models produced NSE values
exceeding 0.5 (the acceptance threshold) at most locations (29 and 30 locations respectively). In contrast, the G model failed
to meet the acceptable NSE threshold of 0.5 at 19/32 locations. Furthermore, when analyzing streamflow events with peak
flows exceeding the 95th percentile at each station, the average NSE values for the S and R models consistently outperformed
the G model at 26 and 27 locations respectively, contradicting previous conclusions that training with data from a larger
number of basins should enhance predictions of extreme event. Similar patterns were observed in the LSTM applications for
PUB, where the G model showed relatively poorer performance as compared to the R model. Most notably, when comparing
against the R model within the Great Britain area, the G model failed to provide superior results for any gage location (Fig.
1d).
Based on the analysis of these published model results, and despite the limited number of overlapped basins, we can conclude
that *training LSTM models with data sets from a larger number of basins does not necessarily enhance model performance.*

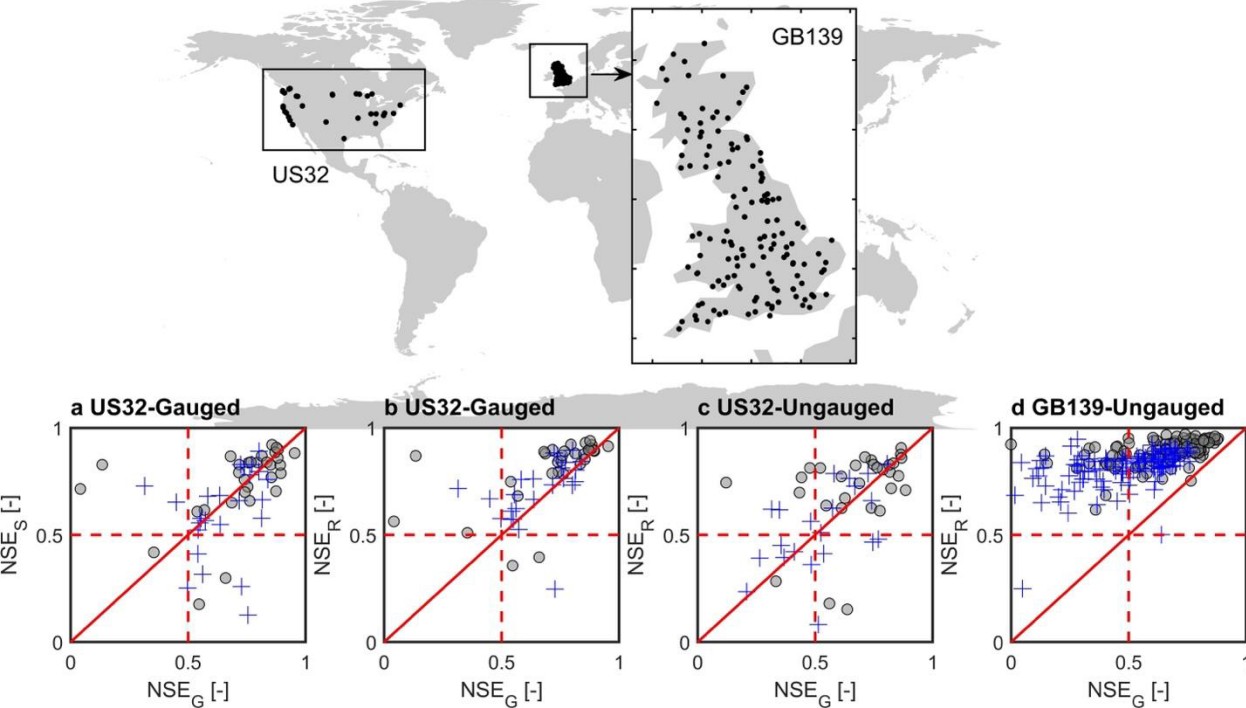

**Figure 1. A performance comparison of models trained using data on global (G) basins versus models trained using data for regional (R) and single (S) basins.** The scatter plots show a performance comparison based on Nash-Sutcliffe Efficiency (NSE) at overlapping gage locations between the G, R, and S models. Gray circles represent NSE values for overall simulation, while blue plus symbols indicate the median NSE values calculated for high flow events with peak flows exceeding the 95th percentile of the entire time series at each forecast location. The number of overlapping gages between the G model and R model for the contiguous United States (CONUS) is 32 stations (USE32-Gauged), while the data set for Great Britain has 139 overlapping stations (GB139-Unaguged). Subplots (a) and (b) present comparison results for forecasting applications in gauged basins. Subplot (a) compares the G and S models, and subplot (b) compares the G and R models. Subplots (c) and (d) show the results for ungauged basins. Red dashed lines represent the NSE threshold of 0.5, indicating the minimum acceptable model performance level. All simulation results were collected from previous studies.

## 4 Does Multiple-Basin Training Consistently Outperform Single-Basin Approaches? Insights from Experimental Studies

To provide a more comprehensive evaluation and ensure fair comparison, as well as to test whether training ML models with multiple basins consistently outperforms the single-basin training approach, we designed and implemented individual S models for 609 globally distributed gauge locations and compared these trained models against the G model. These 609 locations were identified through an analysis of available observed streamflow records (particularly for the testing set to compare with the G model for the period after 2014) and an overlap between the published large-sample hydrology dataset (i.e., Caravan (Kratzert et al., 2023)) and G model prediction gages (with distances not exceeding 1 km, as above). The use of this publicly available





comprehensive dataset can facilitate a more straightforward reproduction of this study's results. Both S and G models were
using the same climate forcing data sources (i.e., ERA5) to ensure the fairness of comparison. The test period ran from 2014
to 2020, aligned with the G model's test phase and constrained by the Caravan dataset's available data. Also, since streamflow
records from several stations in the Caravan dataset only extended until 2014, we merged these data with records from the
Global Runoff Data Center (GRDC) dataset to extend the streamflow time series to 2020. To ensure data quality, we only
included stations where the observed streamflow data between the Caravan and GRDC datasets showed a correlation (based
on $R^2$) exceeding 0.99 (a threshold commonly used for streamflow data processing studies (Do et al., 2018; Gudmundsson and
Seneviratne, 2016).

**Figure 2. Performance comparison between the machine learning model trained using streamflow records for global basins (G**
**model) by Google and six models trained using data from single basins (S models).** Scatter plots (a-f) show the 1:1 comparison of Nash-
Sutcliffe Efficiency (NSE) values calculated based on forecast results for 609 global locations between the G model and each S model. Gray



circles represent NSE values for overall simulation, while blue plus symbols indicate the median NSE values calculated for high flow events
with peak flows exceeding the 95th percentile of the entire time series at each forecast location. Red dashed lines represent the NSE threshold
of 0.5, indicating the minimum acceptable model performance level. Cumulative Distribution Functions (CDF) plots (g) present NSE values
calculated based on the simulations from each model for each forecast location. Subplot (h) shows the CDF of median NSE values calculated
for high flow events. The vertical gray dashed lines indicate performance thresholds where NSE < 0.5 is unsatisfactory, [0.5-0.65] is
acceptable, [0.65-0.75] is good, and [0.75-1] is very good. Simulation results for the G model were collected from previous studies, while
results for six S models were performed from our experiments.
Six different approaches for training S models (called S-1 to S-6 models hereafter) were implemented, with detailed
descriptions provided in Appendix A. Fig. 2 (and Table B.2) illustrates the 1:1 comparison of NSE values between the G model
and each S model variant. Overall, the results demonstrate that the G model does not show clear superiority compared to the
S models, particularly S-3, S-4, and S-6. Specifically, the G model provides better NSE values than these three models at
60.7%, 64.9%, and 18.9% of prediction locations, respectively, while at the remaining significant portion (39.3%, 35.1%, and
81.1%, respectively), the S models show superior performance. When we only consider basins with acceptable model
performance (with NSE > 0.5), the G model has 68.5% of gage locations meeting this requirement. In contrast, S-4 has 68.8%
locations while S-6 models achieve 84.7% (Fig. 3g). When increasing the NSE threshold value, the G model shows better
performance compared to S-4; however, the number of basins is minimal. When considering the same condition of not using
observations as input (as with S-6), analysis of the best-performing model selection between G and the five S models (S-1 to
S-5) reveals that the G model performs best at 329 prediction locations, while S models demonstrate superior performance at
280 locations (representing 46%) out of 609 total locations. These findings further emphasize that the implementation of the
G model does not consistently outperform the use of S models. It's worth noting that our S models were not optimized, as their
hyperparameters were chosen arbitrarily (see Appendix A).



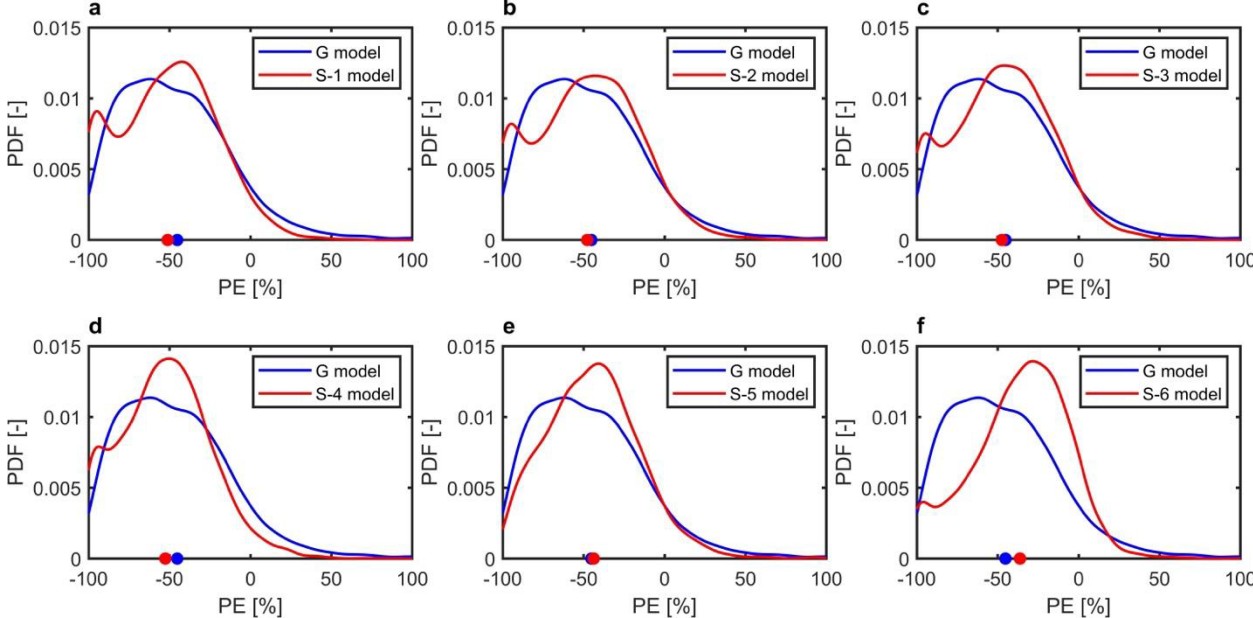


**Figure 3. Performance comparison between the G model and six S models based on the peak error (PE) metrics for high flow events.**
The PDF plots show the distribution PE calculated for individual events across 609 forecast locations. These events were selected based on
their peak flows exceeding the 99th percentile of the total streamflow series for a given gage. While the PE values range from -100% to
infinity, the visualization is focused on the range of -100% to 100%. Blue and red circles denote the mean of PE for G and S models,
respectively.

One key argument was that using data records from multiple basins would improve predictive skill for extreme events – this
claim was supported by evidence from applications using the CAMELS dataset. However, our analysis of the G model's
performance across 5,357 high flow events over 609 locations with peak flows exceeding the 99th percentile of the total
streamflow series (for each location) reveals a different outcome. The PE assessment indicates that the G model substantially
underestimates peak flows, with a median PE value of -45.2%. Comparative analysis with S models (S-1 through S-5) that do
not incorporate data integration shows minimal differentiation, as these models exhibit similar PE values ranging from -52.6%
to -43.7% (Fig. 3). This finding challenges the assertion that multiple-basin training necessarily leads to improved extreme
event predictions.

**5 Discussion**

Many will argue that comparing S, R, and G models in Section 2 is unfair due to their use of different forcing data. The G
model uses global ERA5, while Model R (a regional LSTM) is trained with high-resolution/quality reanalysis data only
available in the US (Daymet + Maurer + NLDAS) or Great Britain (CEH-GEAR + CHESS-PE + CHESS-met), and S model
from Kratzert et al. (2024) uses single-basin training with Daymet + Maurer + NLDAS. It should be noted that LSTM





performance is inherently influenced by forcing data quality, where lower quality forcing data can lead to worse LSTM
performance (Klotz et al., 2025). However, this highlights a critical point: high accuracy forcing data is not universally
available at large scales, particularly globally. This limitation not only constrains model performance but also prevents trained
models from accurately capturing the streamflow generation mechanisms. In other words, training with multiple basins
(especially at a global scale) necessarily involves a trade-off between using high quality data and employing high bias datasets
available across all basins.
Furthermore, beyond hydrometeorological data, numerous studies have demonstrated improved LSTM performance by
incorporating basin-specific data such as reservoir related data or operations (Lang et al., 2025; Kwon et al., 2023). However,
such specialized data is currently unavailable in any large-scale datasets. This limitation can be considered a primary constraint
hindering the training of LSTM with multiple basins. While we acknowledge that training S, R, and G models with uniformly
high-quality data would ensure a more equitable comparison, such an approach is currently infeasible, particularly for global
scale. Our results demonstrate that using multiple-basin training approach with global forcing data (such as ERA5) is not a
solution that guarantees either improved accuracy compared to S or R models trained with high-quality datasets or practical
applicability.
In cases where data availability is limited in a specific region (both in terms of data types and temporal coverage), we believe
that training with multiple-basins and large-scale available dataset represents a valuable solution for extending the training
dataset's temporal coverage. However, in the contrasting scenario, where regions possess diverse and high-quality data types
recorded over several decades, thus, a fundamental question that should be addressed is: *Which approach yields superior
accuracy—training a local model (single basin) with basin-specific data and local-knowledge, or training a model with
multiple basins using commonly available non-local data? Is the trade-off between data quality and data quantity worth it?*

## 6 Conclusion and Outlooks

The key takeaway from our analysis is that training LSTM for rainfall-runoff modeling does not necessarily require data from
multiple, as many as possible basins. Evidently, the training approach should not be used as the standard for model evaluation;
rather, the final performance achieved by the model should be the decisive factor. Even models trained with limited data (i.e.,
for a single basin) can achieve good performance, thus negating the necessity for large-scale training that demands greater
computational resources that aren't readily available to all individuals or organizations worldwide. As demonstrated in this
study, our S models outperformed Google's state-of-the-art ML model trained with nearly 6,000 basins globally, with superior
simulation results at 46% of locations.
While we acknowledge the importance of large-sample hydrology datasets, most pre-processed datasets remain limited,
primarily providing only streamflow data, climate forcings, or static basin attributes. However, effective large-scale streamflow
simulation requires more comprehensive and reliable data. For example, in these datasets, streamflow data are often the only



measured data, while other data types, especially model forcing data (particularly precipitation), are extracted from reanalysis
data sources (e.g., ERA5 in CARAVAN dataset). These data sources contain considerable uncertainty, and the accuracy of
these data varies across different river basins. Also, numerous studies indicate that reservoir operation data plays a crucial role
in providing accurate forecasts for reservoir-influenced river basins operations (Lang et al., 2025; Kwon et al., 2023). This
raises questions about the benefits of multiple-basin training approaches for ML models, when they are applied to such complex
watersheds. Although numerous studies indicate that ML can forecast for such human-impacted watersheds based solely on
forcing data, not all case studies demonstrate successful outcomes.
In conclusion, for applications of streamflow prediction in gaged basins, LSTM model should be customized and optimized
for each unique case study. Training models with a large number of datasets remains one of several good options for model
benchmarking. Most importantly, the training approach should not be the criterion for judging whether a study has "made a
mistake"; instead, the trained model performance should be the determining factor.

**Appendix A: Design of experiments**

**A.1 Dataset**

The G model simulation data, trained by Google across 5,680 stations for the period 1984-2021, were obtained from the public
repository (https://doi.org/10.5281/zenodo.10397664). Within this dataset, simulated data for ungauged basins were extracted
from the "kfold_splits" folder, while simulations for gauged basins were sourced from the "full_run" folder. Here, we utilize
retrospective (or reanalysis) simulation results (at lead time of 0 day) forced by ERA5 data rather than hindcast results using
forcing from other sources. Note that the model outputs are daily and right-labeled timestamps, therefore, we shifted all the
simulations back one day to align with other data using left-labeled timestamps.
The simulations for 531 gauged basins in CONUS using S and R models were collected from Kratzert et al. (2024). The
simulation for ungauged basins using R models was collected from Kratzert et al. (2019). The simulations for 669 GB ungauged
basins were collected from Lees et al. (2021).
Daily climate forcings were obtained from the Caravan dataset (accessed March 23, 2023) (Kratzert et al., 2023), which
provides processed data derived from the ERA5-Land dataset. The forcing variables utilized for ML training included total
precipitation, 2-meter temperature, surface net solar radiation, surface net thermal radiation, snow depth water equivalent, and
surface pressure. Daily observed streamflow (with a unit of cms) data were acquired from the GRDC
(https://grdc.bafg.de/GRDC/, accessed on Augst 1st, 2024), encompassing 10,826 stations, with individual station records
stored in separate text files. All streamflow data from both Caravan and GRDC are standardized in units of mm/day with all
negative values set to NaN.

**A.2 Training strategies**

From our comprehensive literature review, we randomly selected six approaches to train S models. Although the literature
presents numerous effective training methodologies capable of producing robust simulation results, this study does not seek to



identify the superior training practice. Instead, our investigation focuses specifically on comparing the performance of S
models against the state-of-the-art G model developed by Google. Six approaches to train S model include:

- S-1: Model trained using climate forcings as input, with observed streamflow as output. This approach serves as the baseline model, following a common approach employed in (Kratzert et al., 2018).
- S-2: The model was trained using input data that was processed by the discrete wavelet transform (Daubechies function, level 3), with the observed streamflow as the output. This approach leverages the advantages of wavelet transform by decomposing the time series into multiple subseries of lower resolution, and extracting non-trivial and potentially useful information from the original data (Nourani et al., 2014). This method has been extensively employed to solve problems related to the diagnosis, classification, and forecasting of extreme weather events (Nourani et al., 2014; Tran et al., 2021).
- S-3: The model was trained using a combination of forcing data and simulated streamflow from a simple conceptual rainfall-runoff model (SIMHYD (Chiew et al., 2002)) as inputs, with the observed streamflow as the target output. The SIMHYD model was optimized using the Shuffled Complex Evolution - University of Arizona (SCE-UA) method over the period 1980-2013 (Duan et al., 1994). This approach can be considered a hybrid or post-processing method of the SIMHYD forecast results. This approach has also been demonstrated to provide better performance compared to the baseline model (Tang et al., 2023).
- S-4: The model was trained using climate forcings and extended climate forcings randomly selected from other basins as inputs. The training target data combined observed streamflow and SIMHYD simulations using the extended climate forcings. The SIMHYD model was optimized in a similar manner as in the previous approach (S-3). This approach aims to extend the available data for training the model. In cases where the observed streamflow data is limited, an LSTM model can be trained based on the output of the SIMHYD model, meaning the LSTM will mimic the performance of SIMHYD. This trained model is then referred to as a surrogate model (Tran et al., 2020; Tran et al., 2023b). With this approach, there is no need to worry about the lack of data for training the machine learning model. However, the final performance of the surrogate model depends on the performance of the SIMHYD model, as the machine learning model can only approximate the streamflow as well as the SIMHYD model.
- S-5: The model was trained exclusively using data from high flow events with peak flows exceeding the 90th percentile of the streamflow series. Climate forcings served as the input, with the observed streamflow as the target output. This approach aims to minimize the proportion of moderate and low flow data in the training dataset. It is noteworthy that machine learning models typically learn better in the regions where the majority of the data resides, which, in this case, would be the moderate and low flow regions. As a result, these models often treat extreme values as outliers, leading to poor performance in capturing the high flow events. Therefore, this approach reduces the proportion of moderate and low flow data in order to balance the data distribution, with the goal of enabling the machine learning model to learn and recognize the high flow events better, thereby improving its ability to forecast these high flow events accurately (Tran et al., 2023a).
- S-6: The model was trained using climate forcing and observed streamflow data, combined with a data integration approach (Feng et al., 2020; Tran et al., 2021). This means that the observed streamflow data from the preceding time step was used as input to forecast the subsequent time period. This approach has been highlighted in multiple studies for its ability to produce highly accurate simulation results, significantly outperforming the baseline models.

**A.3 LSTM configuration**
All S models used identical hyperparameters, which were chosen arbitrarily rather than through optimization. We acknowledge
that hyperparameter tuning could improve model performance. In other words, this study does not represent the best possible
versions of the S models. Specifically, the architecture utilized a hidden size of 256 with two LSTM layers, a dropout rate of
0.4, sequence length of 365, learning rate of $1\times10^{-4}$, and 100 epochs. The NSE loss function was employed with an Adam



optimizer. Using data spanning from 1981-2020, the model training and validation periods were set to 1981-2004 and 2005-
2013, respectively. The test period, which was used for comparison with the G model, covered 2014-2020.
Initially, a total of 645 locations were identified as overlapping between the Caravan and GRDC datasets. After data
consolidation, many stations showed significant data missing, especially after 2014. We trained LSTM models using 6 different
approaches for all 645 basins automatically. The evaluation results were obtained from a test set corresponding to each basin
and approach. Note that the different training approaches proposed required varying quantities and qualities of input data. A
model was considered successfully trained and applied when it achieved non-NaN NSE values. NSE values were NaN due to:
1) lack of data in either the training set or validation set during the training process, and/or 2) absence of observed data in the
test set. After removing all NaN values of NSE (one for each approach and basin), consequently, consequently, these stations
were excluded from S model training. As a result, the number of models trained for each approach were 606, 606, 606, 609,
606, and 530, respectively. Notably, the S-4 model was trained using SIMHYD model simulations as target output, which
proved resilient to missing data in the 1981-2004 period.

**A.4 Evaluation metrics**

We assess model performance against observation using standard quantitative verification metrics commonly used for
streamflow prediction evaluation, including the NSE and PE.

$$\text{NSE} = 1 - \frac{\sum_{i=1}^{T}\left(Q_i^{Obs} - Q_i^{Sim}\right)^2}{\sum_{i=1}^{T}\left(Q_i^{Obs} - \overline{Q^{Obs}}\right)^2} \tag{1}$$

where $Q_i^{Obs}$ and $Q_i^{Sim}$ are observed and simulated streamflow at time $i$, respectively. $\overline{Q^{Obs}}$ is the observed mean. $T$ denotes the
number of data points. The thresholds for NSE values used to evaluate simulation quality in this study are based on widely
accepted considerations in the community. The classification is as follows: NSE < 0.5 is unsatisfactory, [0.5-0.65) is
acceptable, [0.65-0.75) is good, and NSE ≥ 0.75 is very good (Moriasi et al., 2007). Note that all streamflow data used for
NSE calculations were standardized to area-normalized streamflow discharge with units of millimeters per day (mm/day).
PE is computed as:

$$\text{PE} = \frac{Q_{t_{peak}^{Obs}}^{Sim} - Q_{t_{peak}^{Obs}}^{Obs}}{Q_{t_{peak}^{Obs}}^{Obs}} \times 100 \tag{2}$$

where $Q_{t_{peak}^{Obs}}^{Sim}$ and $Q_{t_{peak}^{Obs}}^{Obs}$ are the flow from model predictions and observations at peak times ($t_{peak}^{Obs}$) from observations,
respectively.



## Appendix B. Tables

**Table B.1.** Performance statistics of G, R, and S models presented in Section 2. $NSE_{overall}$ and $NSE_{highflow}$ values are calculated as the median NSE across all prediction locations considering the entire data series and the median NSE calculated for high flow events, respectively.

| | US32 | | | | | GB139 | |
|---|---|---|---|---|---|---|---|
| **Metric** | Gauged | | | Ungauged | | Ungauged | |
| | G | S | R | G | R | G | R |
| $NSE_{overall}$ [-] | 0.76 | 0.81 | 0.87 | 0.70 | 0.77 | 0.69 | 0.90 |
| $NSE_{highflow}$ [-] | 0.57 | 0.56 | 0.67 | 0.36 | 0.42 | 0.41 | 0.81 |

**Table B.2.** Performance statistics of model G and 6 S models presented in Section 3. $NSE_{overall}$ and $NSE_{highflow}$ values are calculated as the median NSE across all prediction locations considering the entire data series and the median NSE calculated for high flow events, respectively. $N_{best}$ indicates the number of basins where the corresponding model performs best. The selection of the best-performing model is only conducted between G and S-1 to S-5 when the models are used under the same condition of not using observed streamflow as input. $PE_{mean}$ is calculated as the mean of PE computed for 5,357 high flow events for each model.

| **Metric** | **G** | **S-1** | **S-2** | **S-3** | **S-4** | **S-5** | **S-6** |
|---|---|---|---|---|---|---|---|
| $NSE_{overall}$ [-] | 0.65 | 0.61 | 0.61 | 0.65 | 0.62 | 0.50 | 0.84 |
| $NSE_{highflow}$ [-] | 0.18 | 0.17 | 0.13 | 0.24 | 0.15 | 0.20 | 0.65 |
| $N_{bas}$ [-] | 329 | 37 | 41 | 118 | 47 | 37 | - |
| $PE_{mean}$ [%] | -45.2 | -51.2 | -47.7 | -47.4 | -52.6 | -43.7 | -36.2 |

## Code and Data availability

The code for reproducing figures and simulation results of 6 S models are available at https://github.com/vinhngoctran/LSTMtraining. Daily reanalysis data for 5,680 stations covering the period from 1984 to 2021 as well as gauge information and basin attributes, can be accessed at https://doi.org/10.5281/zenodo.10397664. Daily observed streamflow data for 10,826 stations, including gauge information, are available from the Global Runoff Data Center (GRDC, https://grdc.bafg.de/GRDC/). The simulations for CONUS are available at https://zenodo.org/records/11247607 and https://www.hydroshare.org/resource/83ea5312635e44dc824eeb99eda12f06. The simulations for GB regions were downloaded from https://zenodo.org/records/4555820. Daily climate forcings were obtained from the Caravan dataset (https://zenodo.org/records/7540792).



## Acknowledgements

V. N.  Tran and V. Y. Ivanov acknowledge the support of the U.S. National Science Foundation CMMI program award # 2053429 and the Department of Defense, Department of the Navy, the Office of Naval Research award # N00014-23-1-2735. J. Kim was supported by the National Research Foundation of Korea (NRF) grant funded by the Korea government (MSIT)(RS-2022-NR070280).

## Author Contributions

V.N.T designed the study, collected the data, processed the data, analyzed the data, and wrote the first draft. All authors participated in the discussion, writing, and finalization of the manuscript.

**Competing interests.** The authors declare no competing interests.

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
