# Peer review of "Technical note: Does Multiple Basin Training Strategy Guarantee"

_EGUsphere, 2025_

## Referee Comment (RC1)

**Review comments (v2) for "Technical note: Does Multiple Basin Training Strategy Guarantee Superior Machine Learning Performance for Streamflow Predictions in Gaged Basins?" by Tran et al.**

Author: Frederik Kratzert

This is my second solicited review of this manuscript. The first review was for a different journal where the manuscript was rejected in the first round. Therefore, I never had the chance to see the replies/thoughts of the authors to my comments.

Compared to the version I previously reviewed, most of the manuscript remains unchanged. And while in the previous review I wasn't sure if much of the manuscript was based on a misunderstanding, as well as mistakes in the data analysis that led the authors to wrong conclusions, I now have to believe that the authors disagree with my comments and stand behind this manuscript. And maybe because of that, I struggled much more with this review than previously.

Looking at the HESS manuscript type definitions, I think paper does not qualify as a technical note, but much rather could be seen as a direct reply to our recently published HESS Opinions paper, called "HESS Opinions: Never train a Long Short-Term Memory (LSTM) network on a single basin" (Kratzert et al, 2024). The core part of that opinion paper can be summarized in two short points:

- If you have multiple basins with streamflow observations and the same kind of input data, you are on average (much) better training a multi-basin than training single basin models.
- LSTMs suffer from a saturation problem, which is especially pronounced in single basin models and to some degree alleviated (for most basins) when training a multi-basin model.

From reading this discussion paper, my feeling is that the authors took the (on purpose provocative) title of our opinion paper a little bit too seriously, especially the word "*Never*" otherwise I have no explanation for this manuscript. So if the sole purpose of the authors is to answer the question in their title, then the answer is "No". This claim was also never made and the question was already answered in the opinion paper that the authors refer to, which however they seem to have missed. For more details on this particular point, see Section 1 of my review.

Unfortunately this is not the only false accusation about conclusions / statements we apparently have made and not the only point where the authors ignored entire sections of our manuscript that do not align with their story.

The most critical point however is the model comparison. Since it seems like the authors try to not only show that ML does not *guarantee superior performance*, again, a claim that was never made, but only that training single basin models is generally not as bad as presented in our opinion paper, the setup of the model comparison is critical.

- The authors try to derive general recommendations from a comparison of models trained with different data qualities (local/global, high/coarse resolution, reanalysis data / real-time data, simulation model vs. operational flood forecasting model), for details see Sect. 2 of my review.
- In another comparison, where some of the models were trained by the author themselves, they effectively compare their models in a *gauged* setting to a multi-basin model in an *ungauged* setting, essentially showing that the multi-basin model is *as good or better ungauged* than their single basin models *gauged.* For details see Sect. 4 of my review.

I am not sure which way I see forward for this manuscript. In any case, the model comparison needs to be corrected to be able to have a scientific debate about "best practices" (L 61).

**1. Does Multiple Basin Training Guarantee Superior Performance?**

The title of this manuscript raises this question which according to the authors is a conclusion of our opinion paper. However, this is wrong, to a degree that I almost wonder if the authors only read the title. Kratzert et al. (2024) have an entire section dedicated to this topic, called "*Are bigger models better everywhere?*" (Sect. 6), stating in the very first sentence that:

"*Even though the best model, on average, is the model trained on all 531 CAMELS basins, it is not the case that the model trained on all 531 CAMELS basins is better in every basin*"

Further there is a dedicated analysis on this topic including Fig 7 (copied here for convenience) that shows that there are a number of basins, where the single basin model outperformed the regional model with statistical significance (first column) and that also smaller multi-basin models have a couple of basins where they are better than the larger model. So to answer the title of this manuscript: No, bigger models do not guarantee superior performance. But as said above, this was also never a claim that was being made. However, to cite further from Kratzert et al. (2024)

"*All models perform worse than the full regional model in more basins than they perform better in.*"

[Figure]

And further,

"*We have not found a way to (reliably) predict which model will perform best in any particular basin. It is not possible to use metrics from the training period or validation period to (reliably) choose the best model in the test period. Additionally, we have tried extensively to construct a separate predictor model that uses catchment attributes and/or hydrological signatures to predict whether one model will perform better or worse than other models in specific basins. We have not been able to construct a model that performs well at this task. Details of these predictability experiments are out of the scope of this paper, but a relevant example was given by Nearing et al. (2024).*"

To conclude: I struggle to understand how the authors thought that anybody ever made the claim that "superior performance" can be guaranteed when training multi-basin models vs. single basin models.

**2. Model comparison**

One main concern I have with this manuscript is the model comparison. I already noted that in my previous review but the authors have largely ignored my comments, as this part of the manuscript is mostly unchanged.

In science, if we want to investigate if A is better than B, then we have to do this in a setting where we can exclude (as much as possible) any other factor that impacts the results of this experiment.

This is not what has happened here. The authors picked simulations from different studies that use different data and train models for different purposes to then base their conclusion on this model comparison.

Taking one step back, the main source for uncertainty/error in the rainfall-runoff models come from the quality of the input data, especially (but not only) the weather input data. Here however, the authors took model simulations from different models in different studies being trained on different data (with different quality and temporal availability, i.e. real-time available or reanalysis) to support the conclusion of their manuscript.

- Model G (Nearing et al. 2024), an operational flood forecasting model, relies on *globally available* forcing data that is *available in real-time*. For the hindcast data, this is CPC precipitation, IMERG precipitation, as well as data taken from historic ECMWF IFS-Forcasts and GraphCast.
- Model R, three different regional LSTMs trained with *high resolution / high quality reanalysis data*, which are only locally available. The different model R's are from
    - Kratzert et al. (2024) for the US using Daymet + Maurer + NLDAS and being trained in a gauged setting.
    - Kratzert et al. (2019) for the US using NLDAS and being trained in an *ungauged* setting.
    - Lees et al. (2021) for Great Britain using CEH-GEAR + CHESS-PE + CHESS-met being trained in a gauged setting. The authors state in L116 that the results from Lees et al. (2021) are for *ungauged* basins, which is wrong. The study by Lees et al. only includes *gauged* experiments.
- Model S from Kratzert et al. 2024, are single basin models being trained Daymet + Maurer + NLDAS (i.e. same as above).

- S1-S6, the only set of models the authors train themselves, using ERA5-Land data and in the case of S-6, even lagged streamflow data as input.

It makes absolutely no sense comparing models forced by different categories of data (in terms of real-time/reanalysis, coarse/fine resolution, global/local) to make a general statement about the best model training setup. And taking it to the extreme, it makes even less sense to compare a global operational flood forecasting model with a focus on ungauged regions to a

single basin model that gets lagged streamflow as input (S-6). Did anyone really think that a global operational flood forecasting model that is purely based on coarse resolution weather data is better than a model that sees yesterday's streamflow as input?

The point being raised by Kratzert et al. (2024) is: If you have multiple basins with the same kind of data, then it is better to train a multi-basin model than individual single basin models. Obviously, using lagged streamflow as input makes even a single basin model better than a global, pure simulation model. The question is: Is a multi-basin model with lagged streamflow as input better on average better than a single basin model with lagged streamflow? Similarly, it is rather obvious that a regional model with regionally available, high resolution reanalysis data is better than another model that was trained on globally (and in real-time) available weather data.

If the authors want to show anything else than that you can find a setting where single basin models are better than multi-basin models, again, something that was already shown in Kratzert et al. (2024) and never claimed to be otherwise, then my recommendation, as in my previous review is the following:

The entire model comparison needs to be done in a controlled environment where all models, multi-basin (be it regional or global) and single basin models, have the same data available. Fun fact: This is exactly the experiment that was done in Kratzert et al. (2024) and the results are known by now.

**3. Sect. 3 "Previous Research Using Single-Basin Trained ML: Was It a Mistake?"**

This entire section is dedicated to the results of the literature review we presented in Sect. 1 of our opinion paper. I left basically the exact same comments in my previous review and they were essentially ignored in this resubmission. Here are a few points the authors make that I think are wrong:

- Having read the studies in question myself, I know that a lot of them (IIRC the majority) use historic streamflow as input. Getting an NSE above 0.75 for an autoregressive model is not a sign of an excellent model, much rather it is relatively simple to achieve, given the high autocorrelation of streamflow in time. For that reason, ad hoc "adequacy" thresholds like NSE above 0.5 or above 0.75 are not relevant for autoregressive models.
- A statement like "*the single basin model is above NSE 0.75*" does not tell you if a different approach (e.g. regional LSTM) would not be significantly better.
- And more general: When performing benchmarking studies and making general statements like the one being done here, judging by a single metric is not really the way forward.

- L 106 "*studies have successfully passed through peer review processes.*" is certainly no criteria for "*no evident flaws in model training approaches*". I don't know who needs to hear this, but this is such a wrong statement that I don't even want to expand on this.

**4. Sect. 4 "Does Multiple-Basin Training Consistently Outperform Single-Basin Approaches? Insights from Experimental"**

This Section has changed to some degree from the previous version of this manuscript. Two things changed:

- The authors trained their single basin models (S1-S6) using Caravan data, which in this case consists of ERA5-Land data.
- The authors corrected an one-off error in their metric computation, which made them underestimate the performance of model G in all evaluations.

However, the following points remain:

- The simulations of model G from Nearing et al. (2024) are from a k-fold cross validation experiment. This effectively means that all predictions for all gauges are from an *ungauged* setting.
- On the other hand, model S1-S6 are naturally models trained in a *gauged* setting.

On top of that, but this only has a minor impact, models S1-S6 use ERA5-Land as input, a forcing product that is not available in real-time and includes data assimilation, a forcing product that is not being used by model G in the operational setting.

**So what the authors effectively show in this section is that model G is as good, or even better (see Table B.2), *ungauged* than all of their single basin models that are not relying on lagged streamflow as input *gauged*. I think this is not what the authors wanted to show but it is actually a remarkable result.**

**5. More equals better!?**

A point that the authors seem to suggest that we made in our opinion paper is that "more data equals better models" and therefore that model G, a global model being trained on thousands of gauges should be better than any of the other models, being trained on just a few hundreds of basins. While the results in Kratzert et al. (2024) are limited to the CAMELS dataset, we have an entire section (Sect. 5 "*Is hydrological diversity always an asset?*") dedicated to this question. This section contains the following sentences:

"*Figure 6 provides evidence that there might be ways to construct training sets that could potentially result in better models than simply training on all available streamflow data. This*

*conclusion is hypothetical because in all the examples shown in Fig. 6, models trained on any subset of the 531 CAMELS basins performed worse, on average, than models trained on all 531 CAMELS basins. However, separating the training set into hydrologically similar groups of basins results in models that perform better than models trained on random basin groups of similar size.*"

So as in some of the points above, I struggle to see where this claim was supposedly made and why the authors don't include this section of our opinion paper into their discussion.

**Line by line comments**

- L28: Funnily enough Kratzert et al. (2018) is probably not the paper you want to cite here, as the LSTM model in that paper was worse than the SAC-SMA in most of the settings. The correct paper to cite here would be Kratzert et al. (2019).
- L 31ff: Most of the points in this paragraph are not any different between ML models and PB models.
    - Where ML people "fine-tune ML architectures and hyperparameters" PB people refine the process implementations in their models and which processes to include.
    - Everybody who works with models constantly "explores alternative training approaches".

- L 51: "*there is no comparison of their model trained using data from multiple basins versus individually trained, basic(sic!)-specific models*", when referring to Kratzert et al. (2024). What do you mean here? The entire opinion is a comparison of a multi-basin model vs individually trained, basin specific models. You even use both types of modeling approaches from that paper in your model comparison.
- L61: There are certainly "best practices". Your title raises the question if multi-basin training can *guarantee* superior performance, and it can't. But if we speak about "best practices" in the average case, then yes, they exist and where shown more than once and Kratzert et al. (2024) is just one example. If you really want to contend the established best practices, then you will have to do model comparison in a controlled setting as described above.
- L68ff: I think a more nuanced view should be considered here.
    - a) Taking the most recent paper from the list of references (Addor et al. 2020), which by now is 5 years old, a significant amount of new data has been made publicly available, covering thousands of stations in tens of countries. This trend, to some degree, has been fueled by the increasing amount of large sample hydrology studies, not exclusively but for sure also including ML applications. While some regions remain white spots on the map, I struggle to see how large-scale ML applications have a detrimental effect on the publication of data,

while I can certainly see how single basin applications would have such a detrimental effect.

- b) If your point is that single-basin approaches have a benefit in regions with only point-based meteorological observations, then you should go ahead and show this, instead of using CAMELS-GB and CAMELS (US).
- L86 Using Knoben et al. (2019) to justify performing model comparison purely based on the Nash-Sutcliff Efficiency almost comical. The last sentence of their paper reads as follows "*More generally, a strong case can be made for moving away from ad hoc use of aggregated efficiency metrics and towards a framework based on purpose-dependent evaluation metrics and benchmarks that allows for more robust model adequacy assessment.*"
- L89 Remove quotation marks around mistakes.
- L89f: Is this list a citation from someone or your own thoughts? If these are references to findings by others, please cite the relevant papers. If they are your own thoughts, please extend further on what you base these statements on. Furthermore, are these unique to ML models? I think all of these points hold for any kind of model and are not specific to ML. Point 1 though is not a mistake I would say, but rather a problem? Because what is the mistake if the trained/calibrated model has poor performance? It is a fact and (hopefully) there is a reason behind it that could be changed to get a better model. But having a model with poor performance is not a *mistake.*
- L 107f: This concluding sentence is wrong on so many levels. I commented above your take of the literature review and certainly what you show here is not contradicting the claim that single-basin training strategy is generally the wrong thing to do.
- L116 as well as Fig 1: The paper by Thomas Lees et al. does not include results for ungauged basins.

**References**

Addor, N., Do, H. X., Alvarez-Garreton, C., Coxon, G., Fowler, K., and Mendoza, P. A.: Large-sample hydrology: recent progress, guidelines for new datasets and grand challenges, Hydrological Sciences Journal, 65, 712-725, 2020.

Knoben, W. J., Freer, J. E., and Woods, R. A.: Inherent benchmark or not? Comparing Nash–Sutcliffe and Kling–Gupta efficiency scores, Hydrology and Earth System Sciences, 23, 4323-4331, 2019

Kratzert, F., Klotz, D., Brenner, C., Schulz, K., and Herrnegger, M.: Rainfall–runoff modelling using long short-term memory (LSTM) networks, Hydrology and Earth System Sciences, 22, 6005-6022, 2018.

Kratzert, F., Klotz, D., Shalev, G., Klambauer, G., Hochreiter, S., and Nearing, G.: Towards learning universal, regional, and local hydrological behaviors via machine learning applied to large-sample datasets, Hydrology and Earth System Sciences, 23, 5089-5110, 2019.

Kratzert, F., Gauch, M., Klotz, D., and Nearing, G.: HESS Opinions: Never train a Long Short-Term Memory (LSTM) network on a single basin, Hydrology and Earth System Sciences, 28, 4187-4201, 2024.

Lees, T., Buechel, M., Anderson, B., Slater, L., Reece, S., Coxon, G., and Dadson, S. J.: Benchmarking data-driven rainfall–runoff models in Great Britain: a comparison of long short-term memory (LSTM)-based models with four lumped conceptual models, Hydrology and Earth System Sciences, 25, 5517-5534, 2021

Nearing, G., Cohen, D., Dube, V., Gauch, M., Gilon, O., Harrigan, S., Hassidim, A., Klotz, D., Kratzert, F., and Metzger, A.: Global prediction of extreme floods in ungauged watersheds, Nature, 627, 559-563, 2024

---

## Author Comment (AC1)

Review comments (v2) for "Technical note: Does Multiple Basin Training Strategy Guarantee Superior Machine Learning Performance for Streamflow Predictions in Gaged Basins?" by Tran et al.

Author: Frederik Kratzert

This is my second solicited review of this manuscript. The first review was for a different journal where the manuscript was rejected in the first round. Therefore, I never had the chance to see the replies/thoughts of the authors to my comments.

Compared to the version I previously reviewed, most of the manuscript remains unchanged. And while in the previous review I wasn't sure if much of the manuscript was based on a misunderstanding, as well as mistakes in the data analysis that led the authors to wrong conclusions, I now have to believe that the authors disagree with my comments and stand behind this manuscript. And maybe because of that, I struggled much more with this review than previously.

Response: We sincerely appreciate the reviewer's continued engagement with our manuscript and recognize the time and effort invested in providing very detailed feedback. We strongly believe in the value of scientific discourse, even when there are apparent disagreements.

With regards to the previous review process, we want to clarify our approach to the manuscript revision. We carefully considered all comments from the previous review and carried out revisions where we found constructive, evidence-based suggestions that would strengthen the material and its scientific contribution. However, we reserved the right to not implement changes for comments based solely on opinions lacking scientific evidence in their support or those that contradicted our empirical findings without specific indication where our results might be incorrect. We inquired the journal to which this manuscript was submitted previously whether review comments and our responses could be shared in the current submission. We were informed this was not permitted under the journal policy: we are not allowed to disclose any information (except the very general information above) about the comments received previously, as this would constitute a breach of research. We have informed the HESS editor regarding this policy of the previous journal.

We note that the reviewer has repeatedly asked the same questions in several places. Therefore, we will only provide detailed responses to these questions the first time they appear. Please find our response to each comment below.

Looking at the HESS manuscript type definitions, I think paper does not qualify as a technical note, but much rather could be seen as a direct reply to our recently published HESS Opinions paper, called "HESS Opinions: Never train a Long Short-Term Memory (LSTM) network on a single basin" (Kratzert et al, 2024). The core part of that opinion paper can be summarized in two short points:

- If you have multiple basins with streamflow observations and the same kind of input
data, you are on average (much) better training a multi-basin than training single basin
models.

- LSTMs suffer from a saturation problem, which is especially pronounced in single
basin models and to some degree alleviated (for most basins) when training a multi-
basin model.

Response: First, regarding the statement "*core parts of that opinion paper that can be*
*summarized*", we are certain that the first statement was not presented in Kratzert et al. (2024)
as we cannot find it, not even a sufficiently close version.

More specifically, Kratzert et al. (2024) do not state anything similar to "*If you have multiple*
*basins with streamflow observations and the same kind of input data*". There is no phrase "*and*
*the same kind of input data*" or an equivalent of that in their conclusions.

Regarding the second point about the LSTM saturation problems, we partially concur that this
issue is "*especially pronounced in single basin models and to some degree alleviated when*
*training a multi-basin model*". However, this conclusion applies primarily when **multiple**
**basins share identical input data quality and types**—a condition that Kratzert et al. (2024) did
not explicitly address.

We discussed this limitation extensively in L216-219 of our manuscript: "*numerous studies*
*have demonstrated improved LSTM performance by incorporating basin-specific data such as*
*reservoir related data or operations (Lang et al., 2025; Kwon et al., 2023). However, such*
*specialized data is currently unavailable in any large-scale datasets. This limitation can be*
*considered a primary constraint hindering the training of LSTM with multiple basins*". We
further noted that "*in cases where data availability is limited in a specific region (both in terms*
*of data types and temporal coverage), training with multiple-basins represents a valuable*
*solution for extending temporal coverage. However, where regions possess diverse and high-*
*quality data types recorded over several decades, a fundamental question emerges: Which*
*approach yields superior accuracy—training a local model with basin-specific data and local*
*knowledge, or training with multiple basins using commonly available non-local data? Is the*
*trade-off between data quality and data quantity worthwhile?*"

**While we acknowledge that multi-basin training can improve model performance**
**compared to single-basin approaches,** as clearly demonstrated in Kratzert et al. (2024), our
findings indicate this does not hold universally across varying datasets. I.e., there is no single
training strategy. As stated in our abstract: "*we compared the G model with our single-basin (S)*
*ML models, trained for 609 global locations individually, and found that the G model does not*
*consistently outperform S models, as results show S models outperforming the G model in 46%*
*of case studies*". This suggests that, given sufficient training data and appropriate model
complexity, the performance "saturation" point of a local model can exceed that of a global
model in a substantial fraction of cases. Therefore, the phrase *"especially pronounced"* (in the
reviewer's comment above) should be used only in specific contexts.

Our study in fact presents the two statements provided by the reviewer. Specifically, the first
statement is echoed in L224-226 (we copied it below) of the original manuscript and the
second one is demonstrated throughout the manuscript (e.g., L53-56, L224-226).

*L53-56: While it is reasonable to assume that multi-basin training could enhance simulation*
*performance through increased training data volume (particularly for extreme events), their*
*conclusions appear to be valid only within a specific research context, specifically, LSTM*
*application to the CAMELS dataset containing 531 nearly natural basins of the contiguous US*
*(CONUS) region.*

*L224-226: "In cases where data availability is limited in a specific region (both in terms of*
*data types and temporal coverage), we believe that training with multiple-basins and large-*
*scale available dataset represents a valuable solution for extending the training dataset's*
*temporal coverage"*

Regarding the comment "*I think paper does not qualify as a technical note*" – we respectfully
disagree with this opinion. Based on the author guidelines from HESS, a technical note is:
"*Technical notes report new developments, significant advances, and novel aspects of*
*experimental and theoretical methods and techniques which are relevant for scientific*
*investigations within the journal scope.*"

Our manuscript indeed reports "*novel aspects of* experimental and theoretical methods and
techniques", specifically regarding training LSTM models. Here we present our findings based
on novel results rather than on an unsubstantiated opinion.

Whether this manuscript should be considered as a technical note or a direct reply to Kratzer et
al. (2024), we have the full trust in the editors of HESS and leave the final decision to them.

From reading this discussion paper, my feeling is that the authors took the (on purpose
provocative) title of our opinion paper a little bit too seriously, especially the word "*Never*"
otherwise I have no explanation for this manuscript. So if the sole purpose of the authors is to
answer the question in their title, then the answer is "No". This claim was also never made and
the question was already answered in the opinion paper that the authors refer to, which however
they seem to have missed. For more details on this particular point, see Section 1 of my review.

Response: When reviewing the work of Kratzer et al., 2024, we carefully apparently checked the
entire manuscript. The main focus of our work is not based on the title, it stems from the two
main conclusions made in Kratzer et al. [2024]. They were stated in L40-49 in the original
version of our manuscript, and are reproduced below:

*"...However, recent discussions have emphasized the need for standardized training protocols,*
*particularly concerning the use of data from either single (individually targeted) or multiple*
*basins for model training. For instance, a recent study put forward two significant assertions*
*stating (Kratzert et al., 2024):*

*- "A large majority of studies that use this type of model do not follow best practices, and there*
*is one mistake in particular that is common: training deep learning models on small,*
*homogeneous data sets, typically data from only a single hydrological basin"*

*- and "LSTM rainfall–runoff models are best when trained with data from a large number of*
*basins".*

where "*this type of model*" in the above quote refers to LSTM models trained using data from a single basin. We contend that both statements are only relevant for specific applications and require further scientific evidence from other studies.

Regarding our manuscript title, thank you for pointing that out. What we discussed goes beyond simply answering such a question. We have revised the title to "I**s *Multiple Basin Training the Best Practice for Machine Learning Streamflow Prediction in Gauged Basins?*"** The purpose of our study is to rigorously address the research question posed in our title by providing a thorough comparison.

Unfortunately this is not the only false accusation about conclusions / statements we apparently have made and not the only point where the authors ignored entire sections of our manuscript that do not align with their story.

Response: We respectively reject the characterization of our analysis as containing "false accusations". Our citations and interpretations of Kratzert et al. (2024) are based on direct quotes from the published text, and if we missed the context, this should be cited in the review.

The most critical point however is the model comparison. Since it seems like the authors try to not only show that ML does not *guarantee superior performance*, again, a claim that was never made, but only that training single basin models is generally not as bad as presented in our opinion paper, the setup of the model comparison is critical.

- The authors try to derive general recommendations from a comparison of models trained with different data qualities (local/global, high/coarse resolution, reanalysis data /real-time data, simulation model vs. operational flood forecasting model), for details see Sect. 2 of my review.

- In another comparison, where some of the models were trained by the author themselves, they effectively compare their models in a *gauged* setting to a multi-basin model in an *ungauged* setting, essentially showing that the multi-basin model is *as good or better ungauged* than their single basin models *gauged*. For details see Sect. 4 of my review.

I am not sure which way I see forward for this manuscript. In any case, the model comparison needs to be corrected to be able to have a scientific debate about "best practices" (L 61).

Response: We confirm that we did not state in our manuscript that *'guarantee superior performance' was stated by Kratzert et al. (2024)*. We believe the reviewer is misrepresenting our work.

Regarding the model comparison criticisms (the first statement), the reviewer characterizes our use of cases with different data qualities as a weakness, but this actually represents the reality of hydrological modeling practice. Real-world applications inherently involve varying data qualities, resolutions, and sources. Our comparison reflects this actual practical reality rather than idealized conditions often assumed in theoretical discussions. This diversity of data availability for model training in fact strengthens, rather than weakens our conclusions about the practical value of different modeling approaches. The ultimate objective in training a model is to have as good and robust model performance as possible. Therefore, there is an apparent reason to (1) train the model with heterogeneous data (different quality, resolution, different hydroclimatic condition) or (2) train with different model architectures and optimization techniques to select the best and most robust model. If we keep training models using data of the same quality – dictated by their global availability, the model architecture, optimization techniques, etc., we might be trapped in a "local minimum" of the model performance. We will miss a chance to have another model that might perform better.

With respect, the reviewer's characterization oversimplifies our analysis. Our comparison demonstrates that single-basin model strategy can achieve competitive performance even when multi-basin model strategy has the theoretical advantage of training on larger datasets – expressed in Kratzer et al. (2024). If anything, this supports the attractiveness of another model training strategy – *if* it can be better.

We disagree that our model comparison needs a "correction". Instead, we believe our approach reflects the complexity and diversity of real-world hydrological modeling challenges better than artificially controlled comparisons. The scientific debate about "best practices" should encompass this real-world complexity rather than being confined to idealized scenarios.

Please find our responses to each comment below.

**1. Does Multiple Basin Training Guarantee Superior Performance?**

The title of this manuscript raises this question which according to the authors is a conclusion of our opinion paper. However, this is wrong, to a degree that I almost wonder if the authors only read the title. Kratzert et al. (2024) have an entire section dedicated to this topic, called "*Are bigger models better everywhere?*" (Sect. 6), stating in the very first sentence that:

"*Even though the best model, on average, is the model trained on all 531 CAMELS basins, it is not the case that the model trained on all 531 CAMELS basins is better in every basin*"

Further there is a dedicated analysis on this topic including Fig 7 (copied here for convenience) that shows that there are a number of basins, where the single basin model outperformed the regional model with statistical significance (first column) and that also smaller multi-basin models have a couple of basins where they are better than the larger model. So to answer the title of this manuscript: No, bigger models do not guarantee superior performance. But as said above, this was also never a claim that was being made. However, to cite further from Kratzert et al. (2024)

"*All models perform worse than the full regional model in more basins than they perform better in.*"

[Figure]

And further,

"*We have not found a way to (reliably) predict which model will perform best in any particular*
*basin. It is not possible to use metrics from the training period or validation period to (reliably)*
*choose the best model in the test period. Additionally, we have tried extensively to construct a*
*separate predictor model that uses catchment attributes and/or hydrological signatures to*
*predict whether one model will perform better or worse than other models in specific basins. We*
*have not been able to construct a model that performs well at this task. Details of these*
*predictability experiments are out of the scope of this paper, but a relevant example was given*
*by Nearing et al. (2024).*"

To conclude: I struggle to understand how the authors thought that anybody ever made the
claim that "superior performance" can be guaranteed when training multi-basin models vs.
single basin models.

Response: We appreciate the reviewer highlighting Section 6 of their paper, as it actually
reinforces our central argument rather than contradicts it. We have revised our manuscript title to
avoid misrepresentation and clarify that our research that addresses two specific questions: (1)
whether training a model with a single basin is a mistake, and (2) whether multi-basin training
represents a universal best practice.

Importantly, Section 6 of Kratzert et al. (2024) does not directly address these fundamental
questions about the validity of single-basin approaches or the establishment of universal best
practices. The evidence the reviewer presents actually supports our position:

- Their own results show single-basin models outperform regional models in specific cases.
This directly falsifies the conclusion that single-basin training is a "mistake" and that
multi-basin approaches are not superior always, although they are "on average".

- Their statement that "*we have not found a way to (reliably) predict which model will*
*perform best in any particular basin*" undermines claims about universal best practices
and supports our argument for methodological diversity.

The contradiction lies not in our interpretation, but in the disconnect between their empirical
findings (which showcase specific advantages for *different* approaches) and their broader claims
about "mistakes" and "best practices". Kratzer et al. (2024) results demonstrate that both
approaches have merit depending on the specific application—**precisely the nuanced view we**
**advocate in this note**. However, these results were not used. Note that these results were based
on only a single input scenario (the models use the same input).

## 2. Model comparison

One main concern I have with this manuscript is the model comparison. I already noted that in
my previous review but the authors have largely ignored my comments, as this part of the
manuscript is mostly unchanged.

In science, if we want to investigate if A is better than B, then we have to do this in a setting
where we can exclude (as much as possible) any other factor that impacts the results of this
experiment.

This is not what has happened here. The authors picked simulations from different studies that
use different data and train models for different purposes to then base their conclusion on this
model comparison.

Response: Since this comment is repeated, please see our response above. Using reviewer's
example above, our main point is that if A is better than B in certain cases, but worse than B in
numerous others, there is no single statement that identifies A as the best. It is more nuanced
than a single best rule.

Additionally, we respectfully disagree that we ignored this question. In this submission, we have
added a discussion section to clarify the rationale for comparing the models.

Taking one step back, the main source for uncertainty/error in the rainfall-runoff models come
from the quality of the input data, especially (but not only) the weather input data. Here
however, the authors took model simulations from different models in different studies being
trained on different data (with different quality and temporal availability, i.e. real-time available
or reanalysis) to support the conclusion of their manuscript.

- Model G (Nearing et al. 2024), an operational flood forecasting model, relies on
*globally available* forcing data that is *available in real-time*. For the hindcast data, this is CPC precipitation, IMERG precipitation, as well as data taken from historic ECMWF
IFS-Forcasts and GraphCast.

Response: It should be clear that we used reanalysis data but not "real-time forecast" data. The
dataset Nearing reported contains reanalysis data that was simulated using forcings considered as
observations rather than using forecast forcings. Additionally, only hindcast simulations with
lead times greater than 1 day used ECMWF IFS-Forecast data – we did not use these hindcast
simulations in our work.

As far as we learned from the publication, forecast data from GraphCast was not used in the G
model as reported in Nearing et al. (2024).

Nearing, Grey, Deborah Cohen, Vusumuzi Dube, Martin Gauch, Oren Gilon, Shaun Harrigan,
Avinatan Hassidim et al. "Global prediction of extreme floods in ungauged
watersheds." *Nature* 627, no. 8004 (2024): 559-563.

Model R, three different regional LSTMs trained with *high resolution / high quality*
*reanalysis data*, which are only locally available. The different model R's are from

-    Kratzert et al. (2024) for the US using Daymet + Maurer + NLDAS and
being trained in a gauged setting.

-    Kratzert et al. (2019) for the US using NLDAS and being trained in an *ungauged*

setting.

-    Lees et al. (2021) for Great Britain using CEH-GEAR + CHESS-PE +

CHESS-met being trained in a gauged setting. The authors state in L116 that the
results from Lees et al. (2021) are for *ungauged* basins, which is wrong. The
study by Lees et al. only includes *gauged* experiments.

-   Model S from Kratzert et al. 2024, are single basin models being trained Daymet +
Maurer + NLDAS (i.e. same as above).

Response: All detailed information on forcings to S and R models was reported in the
Discussion section of the original manuscript (L206-209). Also, based on the
reviewer's clarification, we have corrected the comparison between G and R models
for GB. Here, the simulation results of the G model were used for the gauged
experiment. The results are shown in Fig. R1 below. It can be seen that the R model
developed by Lees et al. (2021) still completely outperforms the G model.

[Figure]

Figure R1. **A performance comparison of models trained using data on global (G) basins versus models trained using data for regional (R) and single (S) basins.** The scatter plots show a performance comparison based on Nash-Sutcliffe Efficiency (NSE) at overlapping gage locations between the G, R, and S models. Gray circles represent NSE values for overall simulation, while blue plus symbols indicate the median NSE values calculated for high flow events with peak flows exceeding the 95th percentile of the entire time series at each forecast location. The number of overlapping gages between the G model and R model for the contiguous United States (CONUS) is 31 stations (US31-Gauged), while the data set for Great Britain has 124 overlapping stations (GB124-Gauged). Subplots (a) and (b) present comparison results for forecasting applications in gauged basins. Subplot (a) compares the G and S models, and subplot (b) compares the G and R models. Subplots (c) and (d) show the results for ungauged basins. Red dashed lines represent the NSE threshold of 0.5, indicating the minimum acceptable model performance level. All simulation results were collected from previous studies.

- S1-S6, the only set of models the authors train themselves, using ERA5-Land data and in the case of S-6, even lagged streamflow data as input.

It makes absolutely no sense comparing models forced by different categories of data (in terms of real-time/reanalysis, coarse/fine resolution, global/local) to make a general statement about the best model training setup. And taking it to the extreme, it makes even less sense to compare a global operational flood forecasting model with a focus on ungauged regions to a single basin model that gets lagged streamflow as input (S-6). Did anyone really think that a global operational flood forecasting model that is purely based on coarse resolution weather data is better than a model that sees yesterday's streamflow as input?

Response: The reason for the model comparison was discussed in the original manuscript. Please find our response below as well.

The point being raised by Kratzert et al. (2024) is: If you have multiple basins with the same kind of data, then it is better to train a multi-basin model than individual single basin models.

Obviously, using lagged streamflow as input makes even a single basin model better than a global, pure simulation model. The question is: Is a multi-basin model with lagged streamflow as input better on average better than a single basin model with lagged streamflow? Similarly, it is rather obvious that a regional model with regionally available, high resolution reanalysis data is better than another model that was trained on globally (and in real-time) available weather data.

Response: Respectfully, the reviewer misrepresents their own study. We could not find any statement similar to "*If you have multiple basins **with the same kind of data**, then it is better to train a multi-basin model than individual single basin models.*" in Kratzert et al. (2024).

Additionally, unfortunately, all the above questions from the reviewer or the reviewer's opinions do not prove why training a model with a single basin is a "mistake" and why training multiple basins is best practice, even though the resultant model does not deliver better results compared to S models or models trained with fewer basins.

The reviewer's comment above also acknowledges that having high resolution/high-quality data will provide advantages in training models. *So if such data are available, why would one need to train a model with coarse resolution data but available for multiple basins?* This is precisely our question and is discussed extensively in this paper. However, what the reviewer comments above was never mentioned previously in Kratzert et al. (2024).

We do not find any specific reason why observed streamflow should not be used as input when it can enhance model performance. Both our work and Kratzert's refer to applications for gauged basins. Therefore, maximizing the use of available data to provide a model with good performance should not be considered a mistake. The point here is that, in hydrology, the ultimate goal is to provide a model with better forecasting capability.

Note also that among the 6 S models we trained, only S-6 uses observed streamflow. Most of our analyses in the manuscript do not emphasize S-6 excessively. Specifically, what we present in the manuscript related to S-6 is in 2 sentences (L179-180 and L183-184) in the original manuscript. Removing S-6 from the experiment would not significantly affect the final findings.

If the authors want to show anything else than that you can find a setting where single basin models are better than multi-basin models, again, something that was already shown in Kratzert et al. (2024) and never claimed to be otherwise, then my recommendation, as in my previous review is the following:

The entire model comparison needs to be done in a controlled environment where all models, multi-basin (be it regional or global) and single basin models, have the same data available. Fun fact: This is exactly the experiment that was done in Kratzert et al. (2024) and the results are known by now.

Response: We respectfully disagree with the reviewer that models need to be compared in a controlled environment. Our "Discussion" is exactly about that. In practice, as demonstrated in WMO (2009), model selection is based on many factors, particularly on model performance. What is the purpose of the trained model: model performance or model training approach? Based on our perspective, how a model is trained is less important than whether the model's performance can be effectively utilized. Best practices are determined by models with the best performance, not by how it was trained.

Additionally, we have carefully read Kratzert et al. (2024) and could not find anywhere stating
that model applications must be conducted in a "*controlled environment where all models, multi-*
*basin (be it regional or global) and single basin models, have the same data available.*"
Although we agree that in hypothesis testing, this is important. However, in studies with high
practical applicability and applications for specific areas (where authors/researchers worldwide
are typically funded), such studies need to have practical significance and real applicability. The
final goal of science is practical application - that is what matters.

Note that the G model was trained with many basins, yet it is clear that the practical value of this
model is questionable when its performance is low. This is especially true in flood forecasting
when it significantly underestimates flood peaks by up to 50%. This raises the question: what is
the purpose of training and operating such a model in practice when the simulation results are
unreliable?

Since this comment is an opinion, we did not make any edits in our manuscript.

WMO (World Meteorological Organization). Guide to hydrological practices, volume II
management of water resources and application of hydrological practices. Geneva,
Switzerland: World Meteorological Organization, 2009.

**3. Sect. 3 "Previous Research Using Single-Basin Trained ML: Was It a Mistake?"**

This entire section is dedicated to the results of the literature review we presented in Sect. 1 of
our opinion paper. I left basically the exact same comments in my previous review and they
were essentially ignored in this resubmission. Here are a few points the authors make that I
think are wrong:

-   Having read the studies in question myself, I know that a lot of them (IIRC the majority)
use historic streamflow as input. Getting an NSE above 0.75 for an autoregressive model
is not a sign of an excellent model, much rather it is relatively simple to achieve, given
the high autocorrelation of streamflow in time. For that reason, ad hoc "adequacy"
thresholds like NSE above 0.5 or above 0.75 are not relevant for autoregressive models.

Response: First, the use of NSE with a threshold of 0.75 is widely favored in the hydrological
community to assess whether a model has a good performance, regardless of what is "under the
hood". Second, we do not see that using historic streamflow as input is a mistake or problematic
in any way. Note that both Kratzert et al. (2024) and our study refer to training models for
forecasting in gauged basins. Specifically, Kratzert et al. (2024) stated that "*Note that this
analysis does not account for the value of hydrologic diversity for prediction in ungauged
basins.*" The use of historical data (for data assimilation) is nearly standard in operational
forecasting systems. This means it is applied in cases where data are available. We do not find
any reason to exclude observed data when it improves model performance.

-   A statement like "*the single basin model is above NSE 0.75*" does not tell you if
a different approach (e.g. regional LSTM) would not be significantly better.

Response: We confirm that we never made this statement in our manuscript.

And more general: When performing benchmarking studies and making general
statements like the one being done here, judging by a single metric is not really the way
forward.

Response: We partially agree with the reviewer that the threshold values of NSE are somewhat
subjective and one-sided. However, these thresholds have been widely used (and have
essentially become standard) in hydrological science. Therefore, we are simply following the
common standards widely recognized by the community. Additionally, NSE is reported in
most of the 109 studies we have referenced. Therefore, we believe there is no issue with using
NSE to evaluate model performance.

Furthermore, while the reviewer stated that "I*f you have multiple basins with streamflow*
*observations and the same kind of input data, you are on average (much) better training a*
*multi-basin than training single basin models*", many of the studies we reviewed do not meet
such data requirements, as numerous studies use local data with limited basin coverage. We
can easily find that Kratzert et al. (2024) criticized these studies and characterized the single-
basin training approach as a "mistake". However, no evidence was presented in Kratzert et al.
(2024) to support this claim. In this study, we carefully reviewed these studies and posed a
question that remains unanswered: why should a model with good performance be considered a
"mistake"?

-   L 106 "*studies have successfully passed through peer review processes.*" is certainly no
criteria for "*no evident flaws in model training approaches*". I don't know who needs to
hear this, but this is such a wrong statement that I don't even want to expand on this.

Response: Thanks for the comment. We have removed this statement to avoid the confusion.

**4.  Sect. 4 "Does Multiple-Basin Training Consistently Outperform Single-Basin**
**Approaches? Insights from Experimental"**

This Section has changed to some degree from the previous version of this manuscript. Two
things changed:

-   The authors trained their single basin models (S1-S6) using Caravan data, which in this
case consists of ERA5-Land data.

-   The authors corrected an one-off error in their metric computation, which made them
underestimate the performance of model G in all evaluations.

However, the following points remain:

-   The simulations of model G from Nearing et al. (2024) are from a k-fold cross
validation experiment. This effectively means that all predictions for all gauges are
from an *ungauged* setting.

-   On the other hand, model S1-S6 are naturally models trained in a *gauged* setting.

Response: We confirm that we used data from the full_run folder (containing data for gauged
setting) but not using data from the kfold folder (containing simulation data for ungauged
setting). This has been stated in L255-256 of the original manuscript.

On top of that, but this only has a minor impact, models S1-S6 use ERA5-Land as input, a forcing product that is not available in real-time and includes data assimilation, a forcing product that is not being used by model G in the operational setting.

Response: We have mentioned above that the simulation data from the G model is reanalysis data (with lead-time of zero day) with inputs including ERA5-Land reanalysis, CPC, and IMERG, but not with forecast forcings from IFS (for 1-7 days lead-time predictions).

**So what the authors effectively show in this section is that model G is as good, or even better (see Table B.2), *ungauged* than all of their single basin models that are not relying on lagged streamflow as input *gauged*. I think this is not what the authors wanted to show but it is actually a remarkable result.**

Response: We believe that the reviewer misunderstood the results. Here, we used the data from the folder "full_run" but not " kfold_splits". The simulation results in the fullrun folder are simulations for gauged setting but not ungauged setting. We have provided the data source in the Appendix of the original manuscript.

Again, the data from the G model we used are the reanalysis data (also called lead-time of 0 day) but not forecast (that uses forecast forcing).

Not all six S models used streamflow as input, except for S-6. Our analysis results are based on multiple models, not just S-6. Even with models that do not use observed streamflow as input (such as models S-1-S-5), the performance of S models such as S-3 is equivalent to the G model when based on overall NSE, S-3 is better than the G model based on NSE calculated specifically for high flow, and S-5 is better than the G model specifically for peak flow (based on NSE).

More equals better!?

A point that the authors seem to suggest that we made in our opinion paper is that "more data equals better models" and therefore that model G, a global model being trained on thousands of gauges should be better than any of the other models, being trained on just a few hundreds of basins. While the results in Kratzert et al. (2024) are limited to the CAMELS dataset, we have an entire section (Sect. 5 "*Is hydrological diversity always an asset?*") dedicated to this question. This section contains the following sentences:

"*Figure 6 provides evidence that there might be ways to construct training sets that could potentially result in better models than simply training on all available streamflow data. This conclusion is hypothetical because in all the examples shown in Fig. 6, models trained on any subset of the 531 CAMELS basins performed worse, on average, than models trained on all 531 CAMELS basins. However, separating the training set into hydrologically similar groups of basins results in models that perform better than models trained on random basin groups of similar size.*"

So as in some of the points above, I struggle to see where this claim was supposedly made and why the authors don't include this section of our opinion paper into their discussion.

Response: The reviewer claims they did not advocate for "*more data equals better models*", yet
their own statements clearly contradict this assertion. We provide direct quotes from Kratzert et
al. (2024) below:

In the abstract's final sentence: "*we show that LSTM rainfall–runoff models are best when*
*trained with data from a large number of basins.*"

In the last paragraph of Section 4: " *The blue line in Fig. 6 shows performance (median NSE)*
*increasing as the size of the training data set increases. This effect continues up to the maximum*
*size of the CAMELS data set (531 basins). In other words, it is better to have more basins in the*
*training set, and even these 531 basins are most likely not to be enough to train optimal LSTM*
*models for streamflow.*"

In the first sentence of Section 5's final paragraph: "*The takeaway is that, even if enough basins*
*exist to divide your training data into hydrologically informed training sets, one is likely to be*
*better off simply training a single model with all available data.*"

These statements promote the principle that more data leads to better models. The reviewer's
mention of Section 5 ("*Is hydrological diversity always an asset?*") does not negate this central
thesis—rather, it serves as a brief acknowledgment of potential, while ultimately reinforcing
their primary conclusion that training on all available data is preferable.

Furthermore, the reviewer's own quoted text from Section 5 supports our interpretation: "*models*
*trained on any subset of the 531 CAMELS basins performed worse, on average, than models*
*trained on all 531 CAMELS basins.*" This also supports the "more data equals better models".

We believe that the overall dismissal of single-basin training approaches as "mistakes" lacks
empirical support and contradicts the evidence that well-performing models—regardless of
training methodology—have practical value.

Line by line comments

-   L28: Funnily enough Kratzert et al. (2018) is probably not the paper you want to cite
here, as the LSTM model in that paper was worse than the SAC-SMA in most of the
settings. The correct paper to cite here would be Kratzert et al. (2019).

Response: Thank you for the suggestion. We have updated the reference as recommended.
However, the abstract of Kratzert et al. (2018) explicitly states: "*Using this approach, we*
*were able to achieve better model performance as the SAC-SMA + Snow-17, which*
*underlines the potential of the LSTM for hydrological modelling applications.*"

-   L 31ff: Most of the points in this paragraph are not any different between ML models and
PB models.
-   Where ML people "fine-tune ML architectures and hyperparameters" PB
 people refine the process implementations in their models and which processes
 to include.

- Everybody who works with models constantly "explores alternative
training approaches".

Response: We acknowledge the reviewer's point that both ML and physics-based (PB)
modeling involve optimization and refinement processes. However, we respectfully
disagree that these approaches are equivalent in their fundamental methodologies and
focus.

The key distinction is not in whether both communities engage in model refinement, but
in what they refine and how they approach the modeling problem:

ML approaches are primarily data-driven and focus on optimizing the mapping between
inputs and outputs through architectural choices (network depth, activation functions,
attention mechanisms), hyperparameter tuning (learning rates, batch sizes, regularization),
and data preprocessing strategies. The model learns patterns directly from data without
explicit representation of physical processes.

Physics/process-based approaches, conversely, are theory-driven and focus on refining
the mathematical representation of physical processes (e.g., evapotranspiration, soil
moisture dynamics, routing algorithms). While PB modelers do explore different process
implementations, these choices are constrained by physical understanding and established
equations.

The "alternative training approaches" also differ fundamentally: ML explores different
optimization algorithms, loss functions, and data augmentation techniques, while PB
modeling typically involves parameter calibration within physically meaningful bounds
using methods like Monte Carlo or gradient-based optimization or differentiable learning.

We have revised the paragraph to better clarify these fundamental differences in approach
and philosophy of two model types.

- L 51: "*there is no comparison of their model trained using data from multiple basins*
*versus individually trained, basic(sic!)-specific models*", when referring to Kratzert et
al. (2024). What do you mean here? The entire opinion is a comparison of a multi-
basin model vs individually trained, basin specific models. You even use both types of
modeling approaches from that paper in your model comparison.

Response: We apologize for the typo, it should be "basin-specific models".

Additionally, what we intended to convey was that Kratzert et al. (2024) does not provide a
direct comparison between their multi-basin model and the individually trained, basin-
specific models that they characterized as "mistakes." While their paper does compare
multi-basin versus single-basin approaches in general, they did not specifically evaluate
their proposed model against the existing single-basin studies they criticized, instead
dismissing these approaches without empirical comparison. We have revised the statement
to clarify this distinction and avoid confusion.

*"there is no comparison of their model trained using data from multiple basins versus*
*individually trained, basin-specific models that they characterized as mistakes"*

557 -  L61: There are certainly "best practices". Your title raises the question if multi-basin
558   training can *guarantee* superior performance, and it can't. But if we speak about "best
559   practices" in the average case, then yes, they exist and where shown more than once
560   and Kratzert et al. (2024) is just one example. If you really want to contend the
561   established best practices, then you will have to do model comparison in a controlled
562   setting as described above.

563 Response: We respectfully disagree with the reviewer's characterization of "established
564 best practices" for several fundamental reasons:

565 - The reviewer's claim about "best practices in the average case" is based primarily on
566 Kratzert et al. (2024), which uses data from 531 basins exclusively within the United
567 States. These basins are predominantly near-natural basins that cannot represent the
568 global diversity of hydrological conditions, human impacts, data availability, and
569 modeling challenges that practitioners face worldwide.

570 - What may be statistically superior "on average" in a controlled dataset does not
571 necessarily translate to superior performance in individual real-world applications. The
572 essence of good modeling practice should be achieving reliable predictions for the
573 specific problem at hand, not adherence to a methodology that performs well on average
574 across a particular dataset.

575 - The reviewer appears to base their argument on a narrow subset of studies while
576 ignoring the substantial body of literature we have comprehensively reviewed. Our
577 analysis of 109 studies reveals that many successful applications use single-basin
578 approaches, achieving excellent performance in their specific contexts. Dismissing these
579 as "mistakes" without proper justification contradicts the evidence.

580 - The reviewer seems to focus solely on our title while overlooking our detailed analysis
581 and discussion. We are not arguing against the potential benefits of multi-basin training
582 where applicable, but we rather challenge the overall dismissal of single-basin approaches
583 and the establishment of "best practices" based on limited evidence.

584 - True best practices in hydrology should be context-dependent and evidence-based,
585 considering data availability, basin characteristics, and modeling objectives—not derived
586 from a single methodological preference applied to a geographically limited dataset. This
587 practice was clearly guided in WMO (2009).

588 We maintain that the hydrological modeling community benefits more from
589 methodological diversity and context-appropriate approaches than from rigid adherence to
590 supposedly universal "best practices" based on limited geographical evidence.

591 -  L68ff: I think a more nuanced view should be considered here.

592  -  a) Taking the most recent paper from the list of references (Addor et al. 2020),
593   which by now is 5 years old, a significant amount of new data has been made
594   publicly available, covering thousands of stations in tens of countries. This trend,
595   to some degree, has been fueled by the increasing amount of large sample
596   hydrology studies, not exclusively but for sure also including ML applications.

597   While some regions remain white spots on the map, I struggle to see how

598   large-scale ML applications have a detrimental effect on the publication of data, while I can certainly see how single basin applications would have such a
detrimental effect.

-    b) If your point is that single-basin approaches have a benefit in regions with only
point-based meteorological observations, then you should go ahead and show
this, instead of using CAMELS-GB and CAMELS (US).

Response: Regarding point (a): While we acknowledge the increase in data availability
mentioned by the reviewer, this improvement remains geographically constrained and
insufficient to support the claim of global applicability. The world comprises nearly 200
countries, yet the reviewer references data from approximately 10-20 countries, primarily
concentrated in North America and Europe with high station densities. This spatial bias
fundamentally limits the generalizability of multi-basin approaches to global applications.

More importantly, we believe the reviewer misses our central point. The impact of different
modeling approaches on data publication initiatives lies beyond the scope of our study. Our
emphasis is on model utility and performance: regardless of training methodology, hydrological
models must provide accurate streamflow simulation and forecasting. The primary purpose of
hydrological models is to simulate and predict streamflow, not to serve as instruments for
promoting data publication efforts.

Regarding point (b): We respectfully disagree with the reviewer's assertion that we need to
demonstrate this separately. The evidence already exists within our comprehensive review of
109 studies, where numerous basin-specific approaches have successfully demonstrated their
effectiveness across diverse hydrological conditions and data availability scenarios. These
studies collectively provide substantial evidence that single-basin approaches can achieve
excellent performance when appropriately applied to local conditions.

L86 Using Knoben et al. (2019) to justify performing model comparison purely based on the
Nash-Sutcliff Efficiency almost comical. The last sentence of their paper reads as follows "*More
generally, a strong case can be made for moving away from ad hoc use of aggregated efficiency
metrics and towards a framework based on purpose-dependent evaluation metrics and
benchmarks that allows for more robust model adequacy assessment.*"

Response: We find no issue with using NSE as a primary metric for evaluating model
simulation quality, and the reviewer's characterization is incorrect in their assessment of both
our approach and the implications of the Knoben et al. (2019) citation.

First, NSE remains the most widely used and accepted metric in hydrological modeling,
including in the Kratzert et al. studies. If NSE is adequate for evaluating multi-basin models in
those studies, it should be equally valid for our comparative analysis.

Second, the quote the reviewer provides actually supports our central argument. Knoben et al.
(2019) advocate for "*purpose-dependent evaluation metrics*", which aligns precisely with our
position that model evaluation should focus on performance relative to the intended application,
not on the training methodology employed. The purpose of streamflow modeling is accurate
prediction, and NSE effectively measures this capability.

Third, we are open to conducting additional analyses using alternative metrics if the reviewer
suggests specific ones. However, changing metrics would not alter our fundamental conclusion:
models should be judged by their performance in achieving their intended purpose, not by
conformity to a particular training paradigm.

-    L89 Remove quotation marks around mistakes.

Response: Done.

-    L89f: Is this list a citation from someone or your own thoughts? If these are references to
 findings by others, please cite the relevant papers. If they are your own thoughts, please
 extend further on what you base these statements on. Furthermore, are these unique to
 ML models? I think all of these points hold for any kind of model and are not specific to
 ML. Point 1 though is not a mistake I would say, but rather a problem? Because what is
 the mistake if the trained/calibrated model has poor performance? It is a fact and
 (hopefully) there is a reason behind it that could be changed to get a better model. But
 having a model with poor performance is not a *mistake.*

Response: Thank you, we agree with point 1 that it would not be a mistake if the trained model
has poor performance. Therefore, we have removed that point. We have added references for the
other mistake types.

-    L 107f: This concluding sentence is wrong on so many levels. I commented above your
 take of the literature review and certainly what you show here is not contradicting the
 claim that single-basin training strategy is generally the wrong thing to do.

Response: The statement "wrong on so many levels" referring to our concluding sentence lacks
the specificity needed for constructive scientific discourse.

Our conclusion is based on a rigorous analysis of 109 studies where single-basin ML models
achieved their stated objectives with good performance according to standard hydrological
metrics. The logical foundation of our argument is straightforward: 1) definition of "mistake"
and 2) empirical evidence from literature studies.

-    L116 as well as Fig 1: The paper by Thomas Lees et al. does not include results for
 ungauged basins.

Response: Thanks, we have corrected the figure (See Fig. R1).

References

Addor, N., Do, H. X., Alvarez-Garreton, C., Coxon, G., Fowler, K., and Mendoza, P. A.: Large-
sample hydrology: recent progress, guidelines for new datasets and grand challenges,
Hydrological Sciences Journal, 65, 712-725, 2020.

Knoben, W. J., Freer, J. E., and Woods, R. A.: Inherent benchmark or not? Comparing Nash–
Sutcliffe and Kling–Gupta efficiency scores, Hydrology and Earth System Sciences, 23, 4323-
4331, 2019

Kratzert, F., Klotz, D., Brenner, C., Schulz, K., and Herrnegger, M.: Rainfall–runoff modelling
using long short-term memory (LSTM) networks, Hydrology and Earth System Sciences, 22,
6005-6022, 2018.

Kratzert, F., Klotz, D., Shalev, G., Klambauer, G., Hochreiter, S., and Nearing, G.: Towards
learning universal, regional, and local hydrological behaviors via machine learning applied
to large-sample datasets, Hydrology and Earth System Sciences, 23, 5089-5110, 2019.

Kratzert, F., Gauch, M., Klotz, D., and Nearing, G.: HESS Opinions: Never train a Long Short-
Term Memory (LSTM) network on a single basin, Hydrology and Earth System Sciences, 28,
4187-4201, 2024.

Lees, T., Buechel, M., Anderson, B., Slater, L., Reece, S., Coxon, G., and Dadson, S. J.:
Benchmarking data-driven rainfall–runoff models in Great Britain: a comparison of long short-
term memory (LSTM)-based models with four lumped conceptual models, Hydrology and Earth
System Sciences, 25, 5517-5534, 2021

Nearing, G., Cohen, D., Dube, V., Gauch, M., Gilon, O., Harrigan, S., Hassidim, A., Klotz, D.,
Kratzert, F., and Metzger, A.: Global prediction of extreme floods in ungauged watersheds,
Nature, 627, 559-563, 2024

---

## Author Comment (AC2)

Reviewer 2:

While the authors raise several important and timely questions regarding training strategies in
hydrological machine learning, I believe the current version of the manuscript is affected by
some methodological issues that limit the strength of its conclusions. After reviewing both the
manuscript and Reviewer 1's detailed comments, I respectfully offer the following major and
minor suggestions for improvement.

Response: We thank the reviewer in recognizing the merit of this work and providing
constructive comments. lease find our responses below along with our responses to Reviewer 1.

Major Comments

1.  Model Comparison Framework and Associated Limitations

I share Reviewer 1's concern regarding the comparability of the models evaluated in this study.
The current experimental design involves comparisons among models trained on datasets with
different resolutions, quality levels, and real-time availability. In particular, the contrast between:

•   Model G (a global operational model using coarse, real-time data),

•   Regional models (using high-quality reanalysis data), and

•   Single-basin models with lagged streamflow inputs (S-6), raises concerns about fairness and interpretability. Because these models operate under different
data assumptions, it becomes challenging to isolate the effect of training strategies alone. As
such, the conclusions drawn about the relative performance of global, regional, and single-basin
approaches may be difficult to support in their current form.

Response: We understand the reviewer's concerns about model comparability. We have
discussed the philosophy of comparison of models with different types and quality of inputs in
detail in the Discussion section.
If this were simply a model development study, we would agree that using the same dataset
would ensure fairer comparison – something that Kratzer et al. (2024) already carried out.
However, the focus here is on the ultimate utility of models - specifically, which model provides
better prediction results in real-world scenarios. Considering this, there is no reason not to (1)
train the model with heterogeneous data (different quality, resolution, different hydroclimatic
condition) or (2) train with different model architectures and optimization techniques to select
the *best and most robust model*. If we keep training models using data of the same quality –
dictated by their global availability, the model architecture, optimization techniques, etc., we
might be trapped in a "local minimum" of the model performance. We will miss a chance to have
another model that might perform better.

We have discussed this in the Discussion section and outlined the role of data quality in model
training. The key question is: *what is the purpose of training a model with multiple basins using*
*less reliable data when that model performs poorly compared to a model trained with high-*
*quality (local) data?* It is worth noting that many studies that Kratzert et al. (2024) claim are

"mistakes" actually use high-quality observational data from gauge stations rather than data from
large sample datasets.

Regarding model S-6, which uses observed streamflow as input, it is clearly stated that this
represents model training for gauged regions in both our study and Kratzert et al. (2024).
Therefore, training models with observed streamflow is not problematic when it delivers
effective and high-quality forecasting. We question why one would not utilize available data to
train a better model, instead ignoring observed streamflow and ultimately obtaining a model with
inferior performance. In practice, when observed streamflow data are available, it is preferred to
use it for model training rather than training without it, and we find no issue with this approach.

1.  Alignment Between Research Question and Study Design

The manuscript appears to address two related but distinct questions:

•   Whether models trained on multi-basin data outperform those trained on single-basin
data (as stated in the title), and

•   Whether locally optimized models using basin-specific data can outperform globally
trained models using only limited local information (the implicit research question).

These are both important questions, but they require different analytical frameworks and
modeling assumptions. To strengthen the manuscript, it may be helpful for the authors to clarify
the primary research question and ensure that the experimental design is tailored to directly
address it. Additionally, the first question has already been explored in earlier studies.

Response: Regarding the research questions, we have stated the main research questions in L76-
81 in the manuscript and provided comprehensive approaches to answer them. Specifically, the
two questions are *whether the studies that Kratzert et al. (2024) listed are mistakes when*
*training with single basins*, and second, *whether training with multiple basins results in better*
*model performance.*
To answer the first question, our approach was to review studies using single basin training
approaches to examine the performance of those models as well as the types of data those models
used. For the second question, we performed direct comparisons with a model that the same
research group developed with published results as well as models that we trained. Note that our
trained models still used ERA5-land forcing rather than local data (such as NLDAS-2/Daymet
for CONUS or CEH-GEAR + CHESS-PE + CHESS-met for GB). We believe that when using
local data sources, the performance of single models would be even better. This further
highlights the importance of local/high-quality data in training ML models.
We emphasize that the criteria for evaluating whether a model is good or not should be based on
the model's performance and its intended use but rather than on how the model is configured.
From an application perspective, as long as the model performs well, that is sufficient. We have
clarified these criteria for evaluating model quality in the revised manuscript.
We also concurred with the Reviewer #1 comment that the previous title of our manuscript was
confusing. Therefore, we revised the title to "*Is Multiple Basin Training the Best Practice for*
*Machine Learning Streamflow Prediction in Gauged Basins?*" to avoid misunderstanding,
particularly regarding the first question, as the reviewer pointed out.

1. Constructive Outlook on Global Model Development

Despite the challenges highlighted in this study, I remain optimistic about the ongoing
development of global hydrological models. As data availability and computational methods
continue to improve, I believe there is great potential for globally trained models to better
incorporate local information.

Rather than viewing the current findings as a critique of global approaches, I suggest framing
them as an opportunity to guide future research toward:

● Better integration of local knowledge within global models,

● Scalable methods for collecting and assimilating basin-specific data, and

● Hybrid modeling strategies that combine global generality with local specificity.

This perspective may help position the work within a more forward-looking and solution-
oriented context.

Response: This is an excellent suggestion. We have added some related content to the outlook
regarding the development of global hydrological models:

*"We acknowledge that as data availability and computational methods continue to improve,*
*there is a great potential for globally trained models to better incorporate local information,*
*particularly given the current strong development trend of global hydrological models (Kraft et*
*al., 2021; Müller Schmied et al., 2021; Emerton et al., 2016). However, it is important to*
*recognize that hydrological forecasting results are only meaningful when applied at the local*
*scale, where water resource management and disaster (flood/drought) decisions are made using*
*results at specific locations and specific watersheds, rather than using general global*
*performance. The analysis results in this study demonstrate that state-of-the-art global models*
*exhibit poor performance, especially in flood forecasting with peak errors reaching ~50%,*
*indicating that substantial work remains to be done to improve modeling systems capable of*
*accurate forecasting at the global scale. In this study, we highlight the need for and importance*
*of high-quality data and integration of local knowledge within global models. For forecasting in*
*gauged basins, methods such as data assimilation or data integration are particularly necessary*
*to provide better forecasting results. This suggests the potential for developing hybrid models*
*that can operate at the global scale, while being configured and customized at the local scale."*

Minor Comments

1. Overlap Criterion for Gauges

The criterion used to identify overlapping gauges ("with distances not exceeding 1 km") would
benefit from further justification. Given potential uncertainties in station locations and historical
relocations, a brief explanation or reference supporting this threshold would help strengthen the
methodological transparency.

Response: Thank you. In the revision, we have improved and added additional criteria for
detecting overlapping gauges. Specifically, beyond the distance between gauges, we also evaluate the correlation between observed data from the datasets. If $R^2>0.99$, we identify them as
overlapped gauges. This results in changes showing 31 and 124 overlapped gauges for CONUS
and GB areas, respectively. References for these criteria have also been added in the revised
manuscript.

1.  Peak Flow Threshold Consistency

The thresholds for identifying peak flow events differ between figures (e.g., 95th percentile in
Figures 1 and 2, versus 99th percentile in Figure 3). Clarifying the rationale behind these choices
would improve the consistency and comparability of the analyses.

Response: We have explained the use of different thresholds in the revised manuscript.
Specifically, the results in Figs. 1 and 2 are to demonstrate the model performance in simulating
high flow. Meanwhile, in Fig. 3, we wanted to evaluate the model performance specifically in
simulating extreme events (here we use the 99th percentile to identify these events). We have
revised the figure caption to clarify that.